# Elevated Plasma Immunoglobulin Levels Prior to Heart Transplantation Are Associated with Poor Post-Transplantation Survival

**DOI:** 10.3390/biology12010061

**Published:** 2022-12-30

**Authors:** Patricia van den Hoogen, Manon M. H. Huibers, Floor W. van den Dolder, Roel de Weger, Erica Siera-de Koning, Marish I. F. Oerlemans, Nicolaas de Jonge, Linda W. van Laake, Pieter A. Doevendans, Joost. P. G. Sluijter, Aryan Vink, Saskia C. A. de Jager

**Affiliations:** 1Laboratory for Experimental Cardiology, University Medical Center Utrecht, 3584 CX Utrecht, The Netherlands; 2Circulatory Health Laboratory, Regenerative Medicine Center, Utrecht University, 3584 CX Utrecht, The Netherlands; 3Department of Pathology, Circulatory Health Laboratory, University Medical Center Utrecht, Utrecht University, 3584 CX Utrecht, The Netherlands; 4Department of Genetics, University Medical Center Utrecht, 3584 CX Utrecht, The Netherlands; 5Department of Cardiology, University Medical Center Utrecht, 3584 CX Utrecht, The Netherlands; 6Netherlands Heart Institute (NLHI), 3511 EP Utrecht, The Netherlands; 7Centraal Militair Hospitaal (CMH), 3584 EZ Utrecht, The Netherlands; 8Laboratory of Translational Immunology, University Medical Center Utrecht, 3584 CX Utrecht, The Netherlands

**Keywords:** HTX, immunoglobulin, graft survival, heart failure, cardiac allograft vasculopathy, rejection

## Abstract

**Simple Summary:**

Heart transplantation is the gold standard in selected patients with end-stage heart failure. However, immune responses to the transplanted heart can lead to major complications and graft failure, limiting long-term survival. It has been shown that antibodies targeting the heart are elevated in end-stage heart failure patients and before transplantation of the donor heart. We aimed to determine whether elevated antibody levels before heart transplantation are associated with a poor outcome after transplantation. We measured four general subtypes of antibodies (IgM, IgG1, IgG2, IgG3) in the blood of patients just before the transplantation was performed. We observe that patients with higher levels of IgG1 and IgG2 antibodies tend to reject the transplanted heart more often compared to patients with lower levels of these antibodies. In addition, we observe that IgG1 and IgG2 levels in these patients remain elevated after transplantation. Together our data suggest that a more active adaptive, antibody-mediated, immune system can lead to graft-related complications in cardiac transplant patients. This implies that a subpopulation of heart transplant patients might benefit from expanded immunosuppressive therapy, which includes medications targeting antibody-producing cells.

**Abstract:**

Cardiac allograft vasculopathy (CAV) and antibody-mediated rejection are immune-mediated, long-term complications that jeopardize graft survival after heart transplantation (HTx). Interestingly, increased plasma levels of immunoglobulins have been found in end-stage heart failure (HF) patients prior to HTx. In this study, we aimed to determine whether increased circulating immunoglobulin levels prior to transplantation are associated with poor post-HTx survival. Pre-and post-HTx plasma samples of 36 cardiac transplant recipient patients were used to determine circulating immunoglobulin levels. In addition, epicardial tissue was collected to determine immunoglobulin deposition in cardiac tissue and assess signs and severity of graft rejection. High levels of IgG1 and IgG2 prior to HTx were associated with a shorter survival post-HTx. Immunoglobulin deposition in cardiac tissue was significantly elevated in patients with a survival of less than 3 years. Patients with high plasma IgG levels pre-HTx also had significantly higher plasma levels after HTx. Furthermore, high pre-HTX levels of IgG1 and IgG2 levels were also significantly increased in patients with inflammatory infiltrate in CAV lesions. Altogether the results of this proof-of-concept study suggest that an activated immune response prior to transplantation negatively affects graft survival.

## 1. Introduction

Heart failure (HF) is a complex clinical syndrome in which the heart is functionally and structurally abnormal, which results in reduced cardiac output or increased filling pressures [1]. Approximately 1–2% of people in the Western world are affected by HF and these numbers are expected to increase [2]. Over the past four decades, heart transplantation (HTx) has evolved as the ultimate treatment for end-stage HF [3,4]. Currently, the 1-year survival rate after HTx is 82%; however, long-term survival rates are relatively low (69% after 5 years and 25% after 20 years) [5,6].

One of the main risk factors influencing patient survival after transplantation is cardiac allograft vasculopathy (CAV) [7,8]. CAV is characterized by diffuse concentric intimal thickening of the coronary arteries, leading to ischemic damage to the transplanted heart or sudden cardiac death [7]. CAV can be classified into different histopathological stages: (1) intima consisting of loose connective tissue with inflammatory cells, (2) lesions with smooth muscle cells or myofibroblasts, and (3) fibrotic lesions [9]. These different CAV stages at autopsy are linked to survival time after transplantation [9]. The exact cause of CAV is not fully elucidated yet, but an immune response against the allograft is suggested to be of major importance [10]. Although the production of most cytokines is hampered due to immunosuppressive therapies, the production of antibodies in CAV patients has been described in multiple studies [11,12].

Antibodies against the donor human leukocyte antigen (HLA) are not the only antibodies observed after HTx. Also, antibodies against non-HLA antigens, like angiotensin II type-1 receptor, endothelin receptor type A, and vimentin have been described to be associated with CAV and transplant rejection [13,14,15,16,17,18,19,20]. This suggests that not only a specific immune response against the HLA of the transplanted heart but also against other non-HLA epitopes might contribute to antibody-mediated rejection and CAV. Recently, it has been shown that end-stage HF patients show substantial amounts of myocardial IgG deposits and increased circulating IgG1 and IgG3 levels [21,22]. These findings suggest that already prior to HTx, the adaptive immune system is activated and the presence of antibodies targeting the myocardium can potentially influence post-HTx survival. This phenomenon is also referred to as HLA sensitization, which occurs upon transfusions, and mechanical circulatory support, mostly being LVAD implantation, and is expected to be present in most cardiac transplant recipients [17,23,24,25,26,27]. Different desensitization protocols have been developed, so far with limited efficacy and it has been suggested a more tailored approach for desensitization is needed [17].

In this study, we explored whether pre-HTx antibody levels are associated with post-HTx outcome. We measured immunoglobulin levels at different time points after HTx, in the plasma of the recipient and in post-mortal tissue lysates of the donor heart. We investigated whether there is a correlation between pre-HTx antibody levels and (a) survival post-HTx, (b) and histological CAV stage.

## 2. Materials and Methods

### 2.1. Patient Population

The collection of myocardial tissue and blood was approved by the local medical ethics committee of the University Medical Center Utrecht under protocol 12/387 UNRAVEL and in compliance with the Declaration of Helsinki [28]. Written informed consent for biobanking of tissue samples and blood was obtained prior to transplantation or, in certain cases from before 2010, waived by the ethics committee when informed consent was not possible due to the death of the patient. Patients were included who underwent autopsy after heart transplantation (HTx). Fresh frozen material of the transplanted heart was collected at the time of autopsy, of which 61 patients were selected with coronary artery samples available. Plasma samples were collected prior to transplantation and at different time points post-HTx. Plasma samples stored after the year 2000 were included for immunoglobulin measurements and resulted in 36 patients who met these criteria. Patients were categorized into four quartiles based on post-HTx survival time, including Q1 (0 to 1-month post-HTx), Q2 (1 month to 3 years post-HTx), Q3 (3 years to 10.5 years post-HTx), and Q4 (>10.5 years post-HTx. Clinical characteristics of patients are listed in Appendix A (tissue lysates) and Appendix A (plasma).

### 2.2. Tissue Collection and (Immuno)histochemistry

The tissues collected for this study were processed using a standardized procedure [29]. The procedure includes slicing the heart above the apex and dividing it into 14 sections and collecting the coronary arteries (left anterior descending (LAD), left circumflex (LCX), and right coronary artery (RCA)) [30]. For each patient included in this study, a representative coronary artery with available fresh frozen tissue is selected for tissue lysates. We (and others) have previously studied the consistency of histological CAV stages among many coronary arteries within one patient [9,31]. The coronary arteries from the hearts that were obtained during the autopsy were studied. Consecutive sections of formalin-fixed paraffin-embedded tissue were stained with Haematoxylin and Eosin (H&E) and Elastic van Gieson (EvG) for general morphology and reviewed by a pathologist. To study vessel wall composition, immunostaining for smooth muscle cells (αSMA; Sigma-Aldrich, Merck KGaA, Darmstadt, Germany, 1:32,000) was used. The histological stages of CAV (H-CAV 0–3) were categorized by cellular density and composition of the intimal layer, as described previously [9]. Histological-CAV 1 phenotype is characterized by an intense inflammatory reaction with severe lymphocyte infiltration in the intima, H-CAV2 are lesions with smooth muscle cells or myofibroblasts and H-CAV3 are fibrotic lesions.

### 2.3. Tissue Lysates

All patients whose fresh frozen epicardial tissue was available, including at least 1 of the major coronary arteries were used for tissue lysate antibody analyses (*n* = 61). Tissue lysates of epicardial tissue were made as described previously [32]. Tissue lysates were aliquoted and stored at −80 °C. In the presented data, antibody concentrations were corrected for input of tissue weight and displayed as µg/mL/mg tissue.

### 2.4. Multiplex Immunoassay for Immunoglobulin Detection

According to the preferred method of detection, we used a Luminex immunoassay to measure antibody levels [23]. IgM levels and IgG subclasses (IgG1, IgG2, and IgG3) in tissue lysates and plasma were measured using a Bio-Plex Pro™ Human Isotyping immunoassay 6-plex (Bio-Rad Laboratories, Veenendaal, The Netherlands, 171A3100M) according to manufacturer’s instructions. Plasma and tissue lysate immunoglobulin levels were calculated using internal standards, included in the immunoassay. Proper dilution was tested prior to measuring the entire cohort and tissue lysates were 100× diluted and plasma samples were 40,000× diluted. Immunoglobulin levels were measured using the Luminex LabScan200 with xPONENT software (Luminex, Austin, TX, USA).

### 2.5. Statistical Analysis

The normal distribution of the data was tested using the Kolmogorov–Smirnov test. A one-way ANOVA, two-way ANOVA, or Kruskal–Wallis test, corrected for multiple testing, was used to compare different groups. Correlation was assessed using the Pearson correlation coefficient. All statistics were performed using SPSS (IBM SPSS Statistics 25; IBM Corp, Armonk, NY, USA) and GraphPad Prism (version 7.04). Two-sided *p*-values of *p* < 0.05 were considered significant.

## 3. Results

### 3.1. High Levels of Pre-HTx Circulating Immunoglobulins Are Associated with Shorter Survival after HTx

To investigate whether pre-HTx immunoglobulin levels are associated with post-HTx overall survival, we measured the levels of IgM and IgG1, IgG2, and IgG3, using luminex immunoassay, prior to HTx and compared these levels with survival time post-HTx. High levels of IgG1 and IgG2 correlated significantly with shorter survival post-HTx (Figure 1, for IgG1 r = −0.42, *p* = 0.01; for IgG2 r = −0.43, *p* = 0.01). Levels of IgM and IgG3 showed the same correlation, albeit without reaching statistical significance (IgM r = −0.33, *p* = 0.052; for IgG3 r = −0.29, *p* = 0.1). Since a relatively large proportion of the patients had a survival of less than one month, we performed subgroup analysis in those patients that survived at least one month (*n* = 21) and investigated whether immunoglobulins levels 1-month post-HTx were similarly associated with post-HTx survival. Indeed, increased levels of IgM, IgG2, and IgG3 1-month post-HTx were also associated with limited post-HTx survival (Appendix A, for IgM r = −0.46, *p* = 0.04; IgG2 r = −0.60, *p* = 0.005; IgG3 r = −0.47, *p* = 0.03).

### 3.2. Patients with High Plasma Immunoglobulin Levels Pre-HTx Also Have Relatively High Plasma Immunoglobulin Levels after HTx

To get more insight into the alterations in levels of immunoglobulins after transplantation, we measured the immunoglobulin concentrations at different time points after HTx. IgG1, IgG2, and IgG3 levels in the plasma of patients after HTx significantly reduced 1 month after transplantation compared to pre-HTx levels (Appendix A, for IgG1 *p* = 0.004; IgG2 *p* = 0.003; IgG3 *p* = 0.006). Immunoglobulin levels showed the tendency to rise again 6 months after transplantation, returning to levels similar to pre-HTX levels. To assess whether pre-HTx immunoglobulin levels correlate to the immunoglobulin levels observed post-HTx, patients were subdivided into two groups, based on high pre-HTx immunoglobulin levels (above the mean concentration) and low pre-HTx levels (below the mean concentration). Interestingly, patients with high antibody titers prior to transplantation also showed significantly higher immunoglobulin levels in the first-month post-HTx compared to patients with lower pre-HTx levels (Figure 2A, for IgM *p* < 0.0001; IgG1 *p* = 0.01; IgG2 *p* = 0.02). In addition, the levels of IgG2 and IgG3 remained elevated 6 months after HTx in patients with high pre-HTx levels compared to low pre-HTx levels (IgG2 *p* = 0.003 and IgG3 *p* = 0.0007). This suggests that the immune system of patients with high immunoglobulin levels prior to transplantation and despite the use of immunosuppression remains in a similar degree of heightened activity after transplantation.

### 3.3. Immunoglobulin Tissue Levels in the Donor Heart Are Higher in Patients with Short-Term Survival Compared to Patients with Long-Term Survival

To study whether local immunoglobulin levels in the heart are also associated with survival, we measured immunoglobulin levels in the epicardial tissue of the transplanted donor heart upon autopsy. Interestingly, in patients that died within the first 3 years after HTx, immunoglobulin levels in epicardial tissue of the donor heart were significantly increased compared to patients that died more than 10.5 years post-HTx (Figure 2B, for IgM *p* = 0.004; IgG1 *p* = 0.004; IgG2 *p* = 0.03; IgG3 *p* = 0.0002).

### 3.4. High Levels of IgG in Plasma and Epicardial Tissue Are Related to the Inflammatory CAV Phenotype

Since we observed that levels of IgG1 and IgG2 pre-HTx correlate with post-HTx survival, we hypothesized that elevated immunoglobulin levels in either plasma or tissue lysates might be related to the presence of CAV. An example of histological-CAV phenotype is presented in Figure 3A,B. As already described in previous observations, CAV is associated with poor graft survival. Interestingly, patients who revealed the inflammatory H-CAV 1 at autopsy also showed higher levels of IgG1 or IgG2 in plasma already prior to HTx compared to patients with H-CAV 2 (Figure 3C, for IgG1 *p* = 0.03; IgG2 *p* = 0.03). Consistent with high plasma levels, also locally in the epicardial tissue of the donor heart, immunoglobulin levels were significantly increased in patients with H-CAV 1 compared to patients without H-CAV (CAV-0) (Figure 3D, for IgG1 *p* = 0.03, IgG2 *p* = 0.03, and IgG3 *p* = 0.03).

## 4. Discussion

A strong common denominator both in the progression of HF and post-cardiac transplantation rejection is inflammation [33]. Although inflammation may be beneficial in the early stages of remodeling of the heart, the inflammatory response also has detrimental effects, especially in chronic phases of cardiac remodeling [34,35]. Innate immune cells, particularly macrophages, as well as cells of the adaptive immune system consisting of T and B cells, remain chronically activated and contribute to adverse cardiac remodeling and rejection upon transplantation [22,36,37]. To improve the survival of patients after cardiac transplantation, it is important to understand the pathophysiological mechanism of post-HTx morbidities that affect long-term survival, like CAV and antibody-mediated rejection. Local production of immunoglobulins in the donor heart is a local rejection process within the transplanted heart [10]. Moreover, systemically, high serum levels of immunoglobulins have been described in CAV patients, despite immunosuppressive therapy [38]. Hence, it is known that cardiac antibodies exist in patients with end-stage HF, already prior to transplantation [21], thereby indicating that in some patients high levels of immunoglobulins already exist before transplantation that might affect the post-HTx outcome. Therefore, in this study, we investigated whether antibody levels prior to HTx are associated with post-HTx outcome. We choose to measure general Ig subtypes as this is applicable to the entire patient population, rather than assessing single antigen-specific autoantibodies. In addition, it is part of the standard biomarkers available at the clinical chemistry lab, allowing it to be more easily implemented in routine care for HTx patients.

Though several studies have shown that post-transplantation Ig levels associate with graft survival, we are one of the first to demonstrate a significant correlation between immunoglobulin levels prior to HTx and survival post-HTx, as high levels of IgG1 or IgG2 were associated with a shorter survival time after HTx. Because the patient cohort includes a relatively large group of patients who died within a few days after transplantation the observed association might be influenced by these short-survival patients. Consistently, also in the subgroup analysis of patients that survived at least one month, this correlation was present, as 1-month post-HTx immunoglobulin levels also correlated with survival. This might indicate that high immunoglobulin levels present already before HTx may negatively affect the survival of the cardiac graft. This is in line with the idea of immune sensitization in patients waiting for cardiac transplantation, which was recently highlighted as a major challenge in cardiac transplantation by the American Heart Association and the International Society for Heart and Lung Transplantation [23]. Immune sensitization is believed to occur upon alloantigen exposure during transfusion and mechanical circulatory support. As cardiac transplantation is the final treatment option for heart failure patients it is to be expected all patients have experienced some degree of immune sensitization prior to transplantation. In patients with a stronger immune sensitization and an activated adaptive immune response before transplantation, immune reactions against the donor heart might be more severe, resulting in increased antibody levels and leading to a more rapid dysfunction of the graft.

Next, we studied alterations in levels of circulating immunoglobulins from the time of transplantation until 6 months post-HTx. One month after HTx, antibody levels were significantly lower compared to pre-HTx levels. This is likely secondary to the high dose of immunosuppressive therapy given early after transplantation [39]. This is in line with other studies, demonstrating decreased levels of IgG early after transplantation, thereby increasing the susceptibility to severe infections [40,41]. In addition, if we consider heart failure as a chronic inflammatory condition that is, in part, driven by the diseased heart itself. For instance, it has been suggested that cardiac unloading (i.e., LVAD) reduces inflammation [42,43,44,45]. A similar response upon HTx can be expected and may consequently result in a reduction in (auto)antibodies. Hence, patients with high immunoglobulin levels prior to HTx maintained relatively high levels after transplantation in comparison to patients with low pre-HTx immunoglobulin levels. This observation is in line with the results of a previous study on lung transplant patients [46]. This suggests that once the adaptive immune system is activated pre-HTx, it remains relatively active after HTx despite immunosuppressive therapies.

In addition to plasma, the levels of immunoglobulins were also measured in the epicardial tissue of the donor heart upon autopsy post-HTx. Interestingly, patients who died in the first 3 years post-HTx showed significantly increased levels of immunoglobulins in the transplanted heart compared to patients with a survival of more than 10 years. This observation is in line with the observations in plasma and suggests that an activated adaptive immune system has an adverse effect on graft survival.

To assess whether high levels of immunoglobulins pre-HTx may be associated with a certain phenotype of CAV after transplantation, patients were divided into the different H-CAV categories. Increased pre-HTx plasma levels of IgG1 and IgG2 were found in patients with H-CAV 1, the inflammatory phenotype of CAV suggesting active CAV development. In addition, epicardial tissue of the donor heart showed an increase in IgG1, IgG2, and IgM upon autopsy in this CAV 1 type. Previous studies already showed an association between circulating inflammatory markers, such as C-reactive protein (CRP), and CAV development [21]. Our results suggest that an activated immune system pre-HTx may influence inflammation in the CAV lesion and thereby the more rapid development of these lesions.

### Study Limitations

Our study was primarily designed as a proof-of-concept study and therefore is limited by the relatively small patient group. Therefore, our results need to be confirmed in future studies with larger patient groups. Because our cohort was based on the availability of autopsy, our cohort contained a relatively large percentage of patients with short survival, thereby not reflecting the average survival after transplantation. The cause of CAV development has not been fully clarified. Unfortunately, our study did not allow us to study the mechanistic involvement of autoantibodies in the development of CAV or antibody-mediated rejection, but this is certainly of interest for future studies.

## 5. Conclusions

In conclusion, in this proof-of-concept study, we show that increased immunoglobulin levels prior to heart transplantation are associated with worse survival post-HTx in a single-center cohort of heart transplant recipients. Interestingly, we show that pre-HTx plasma levels of IgG1 and IgG2 are associated with the inflammatory phenotype of CAV. As the exact cause and mechanisms involved in CAV development are currently not known, our findings may instigate new research into the development and treatment of inflammatory CAV. Overall, our data suggest that a heightened immune activation prior to transplantation may negatively impact graft survival after transplantation. This implies that a subpopulation of HTx patients might benefit from expanded immunosuppressive therapy including B cell targeting therapy, like rituximab. To validate the predictive value of pre-HTX Ig levels for post-HTX survival the results should be established in large-scale cohorts.

## Figures and Tables

**Figure 1 biology-12-00061-f001:**
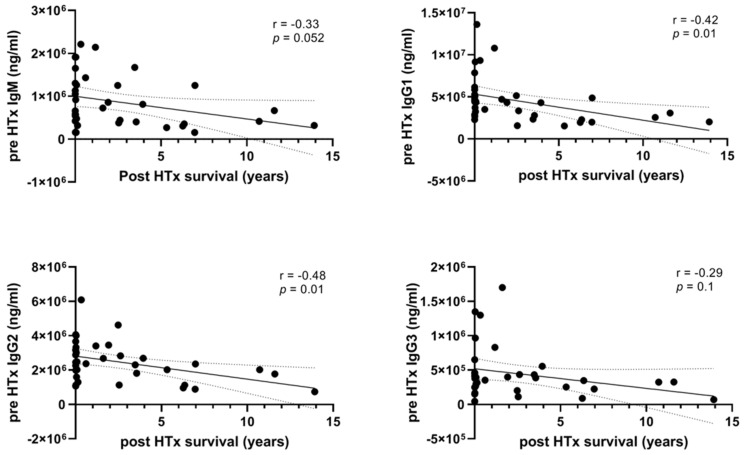
**Pre-HTx circulating immunoglobulin levels are associated with survival time post-HTx.** Pre-HTx plasma levels of different immunoglobulin subtypes were measured using multiplex immunoassay and correlated with survival post-HTx. Levels of IgG1 and IgG2 were significantly correlated with post-HTx survival, where high immunoglobulin levels resulted in a lower survival post-HTx. HTx: heart transplantation. *n* = 36.

**Figure 2 biology-12-00061-f002:**
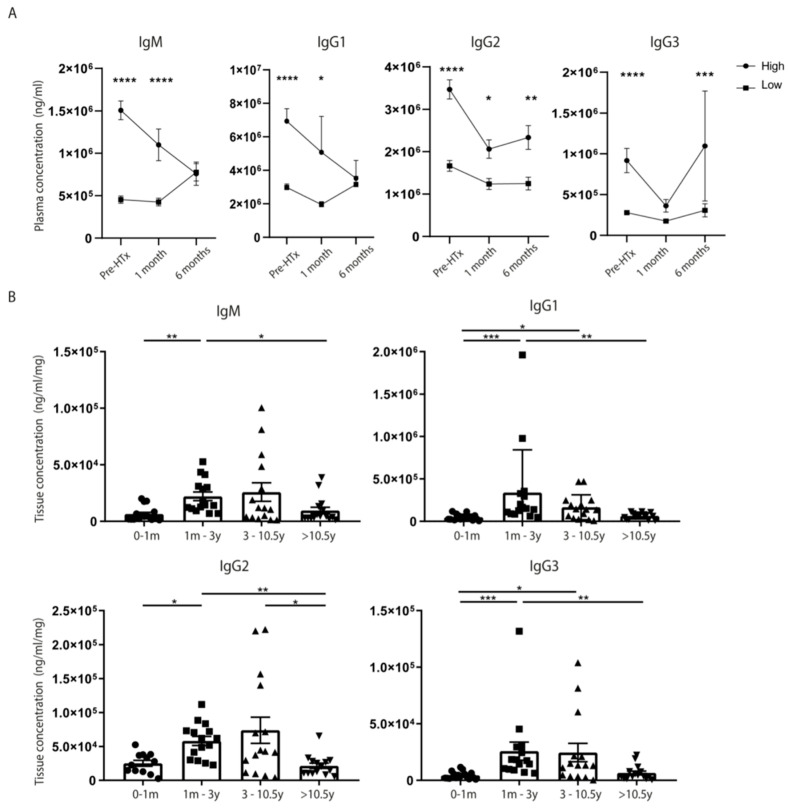
**Immunoglobulin levels in the plasma and tissue pre- and post-HTx.** Plasma immunoglobulin concentrations were measured prior to HTx until 6 months post transplantation. Patients with high antibody titers (above the mean concentration) prior to HTx also remained high in the first 1–6 months compared to patients with low levels (below the mean concentration) before HTx (**A**). In addition, immunoglobulin levels were significantly increased in cardiac tissue of patients who died within the first 3 years post-HTx (**B**). HTx: heart transplantation. For plasma samples: *n* = 36, for tissue lysates *n* = 61 (0–1 month *n* = 16, 1 month–3 years *n* = 15, 3–10.5 years *n* = 15, more than 10.5 years *n* = 15). Data displayed as AVG ± SEM. * *p* < 0.05, ** *p* < 0.01, *** *p* < 0.001, **** *p* < 0.0001.

**Figure 3 biology-12-00061-f003:**
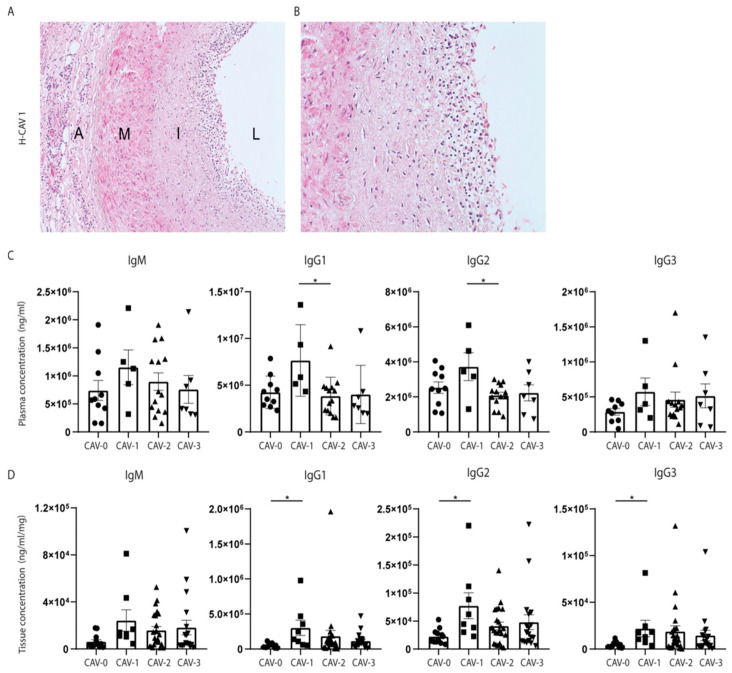
**High pre-HTx plasma levels and post-HTx tissue levels of IgG1 and IgG2 in patients with inflammatory H-CAV 1.** Histology of a coronary artery with H-CAV 1 (**A**), showing the adventitia (A), media (M), and neointima (I) surrounding the lumen (L). High magnification images showing infiltrating lymphocytes in the neointima (A: 10× magnification, B: 20× magnification) (**B**). Patients with H-CAV 1 showed a significant increase in IgG1 and IgG2 plasma levels already prior to HTx compared to patients with H-CAV 2 (**C**). The levels of IgM and IgG3 showed the same trend, although did not reach statistical significance between H-CAV phenotypes. Consistent with plasma levels pre-HTx, the levels of IgG1 and IgG2 were also elevated in cardiac tissue of the transplanted heart with H-CAV 1 (**D**). Data displayed as AVG ± SEM. H-CAV: histological cardiac allograft vasculopathy phenotype, HTx: heart transplantation. For plasma samples H-CAV 0 *n* = 10, H-CAV 1 *n* = 5, H-CAV 2 *n* = 14, H-CAV 3 *n* = 7, for tissue lysates H-CAV 0 *n* = 12, H-CAV 1 *n* = 8, H-CAV 2 *n* = 23, H-CAV 3 *n* = 18. * *p* < 0.05.

## Data Availability

Data are available upon request due to privacy/ethical restrictions.

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
