# Peer review of "Elevated Plasma Immunoglobulin Levels Prior to Heart Transplantation Are Associated with Poor Post-Transplantation Survival"

_biology, 2022, doi:10.3390/biology12010061_

Round 1
Reviewer 1 Report
This paper is well written providing a description of changes in serum and cardiac IgG Abs in pre and post transplant patients. However unfortunately these findings are not entirely new. Patients awaiting heart transplantation shown to have high circulating antibodies to HLA are described by a process called 'sensitisation' associated with CAV, graft rejection and post-transplant mortality. Sensitisation occuring in patients awaiting HTx was recognised in a 2019 Scientific Statement by the AHA (Circ. 2019).
The authors of this manuscript have shown that these antibodies are elevated in heart failure patients and in LVAD awaiting transplantation. This is a valuable finding but again not novel as a variety of autoantibodies targeted for example β1 adrenergic receptors have suggested to play a pivotal role in heart failure. Autoantibodies raised in LVAD patients have also beenassociated with poor outcomes in the acute post transplant stage (JHLT 2009)
It would be valuable therefore if the authors could elaborate on this information and include these previous findings in their work toshow where the novelty of this study lies. The drop in IgG levels at 1 month post-transplantation is valuable and the authors should explore why this occurs in their manuscript. As this is a scientific publication, I also feel that further expanation of why the autoantibodies are raised in the pre-transplant and post transplant ICM/DCM/HCM/RCM patients is strongly warranted.
There are some corrections to make which are otherwise minor but important:
1. Please define the error bars described in figures 2 & 3. Are these representing the SD or S.E.M.?
2. Define HLA as human leukocyte antigens at the first instance.
3. I presume that the local medical ethics committee referred to is the the Biobank Board of the Medical Ethics Committee of the University Medical Centre Utrecht. Please define this in full and that this work was conducted under the following approval (12/387 UNRAVEL Biobank).
4. It would be ideal to define which coronary artery was sampled and at what level. Please define the magnification of the photomicrographs provided in figure 3.
5. Please provide the defining criteria for characterising H-CAV1, 2, & 3 in the Method sections, not in the Results.
6. Please update the references. A number of the ones used are old and there are newer more up to date citations to use.
human leukocyte antigens (HLA). This process by which antibodies are formed is called sensitization.
Reviewer 2 Report
This represents an important study.
The manuscript is very well written and highlights an important topic. It focuses on antibody levels in plasma and transplanted hearts. In doing so, they draw attention to antibody-mediated rejection.
It seems a logical consequence that patients with high immunoglobulin levels in plasma also have increased Ig levels in the tissues. This could only be a consequence of the Ig content in the blood of the vasculature lumen. Authors should discuss this in the manuscript.
The authors did not specify any method (such as complement factor C4d deposition) that would directly indicate antibody-mediated rejection.
The method chosen for antibody measurement in plasma does not seem to be the most appropriate. The authors should at least state in the discussion how their method compares to clinically used methods for measuring immunoglobulins. A dilution factor of 1:40 000 for antibody measurement seems to be very prone to poor reproducibility.
This all needs to be mentioned in a separate paragraph describing the weaknesses of the study. Especially that no C4d has been studies.
Round 2
Reviewer 1 Report
Thank you for the ammendments provided.
Reviewer 2 Report
authors have responded to the questions